# Biomolecular condensates formed by designer minimalistic peptides

Avigail Baruch Leshem[1], Sian Sloan-Dennison[2], Tlalit Massarano[1], Shavit Ben-David[1], Duncan Graham [2], Karen Faulds[2], Hugo E. Gottlieb[3], Jordan H. Chill [3] ✉ & Ayala Lampel[1,4,5,6] ✉

Inspired by the role of intracellular liquid-liquid phase separation (LLPS) in formation of membraneless organelles, there is great interest in developing dynamic compartments formed by LLPS of intrinsically disordered proteins (IDPs) or short peptides. However, the molecular mechanisms underlying the formation of biomolecular condensates have not been fully elucidated, rendering on-demand design of synthetic condensates with tailored physico-chemical functionalities a significant challenge. To address this need, here we design a library of LLPS-promoting peptide building blocks composed of various assembly domains. We show that the LLPS propensity, dynamics, and encapsulation efficiency of compartments can be tuned by changes to the peptide composition. Specifically, with the aid of Raman and NMR spectroscopy, we show that interactions between arginine and aromatic amino acids underlie droplet formation, and that both intra- and intermolecular interactions dictate droplet dynamics. The resulting sequence-structure-function correlation could support the future development of compartments for a variety of applications.

The emerging field of liquid–liquid phase separation (LLPS), as the basis of membraneless organelles formation[1], has triggered a renewed interest in intrinsically disordered proteins (IDPs) and the design of materials based on their remarkable dynamic properties[2]. Membraneless organelles, or biomolecular condensates, are supramolecular disordered compartments that include stress granules, nucleoli, and Cajal bodies. The commonly suggested mechanism for the formation of biomolecular condensates is based on LLPS of IDPs and other biomolecules (mainly nucleic acids)[3,4], in which the condensates' building blocks are highly mobile and exchange rapidly with the surrounding environment. While the exact functionalities of membraneless organelles are still being studied, a general function shared by different biomolecular condensates is concentration, condensation, and storage of proteins, nucleic acids, enzymes and substrates and via this, control of enzymatic reactions and protection of reaction products[4]. Inspired by these remarkable functionalities, researchers have begun to design dynamic compartments that are formed by LLPS[2] of IDPs or polypeptides with disordered domains[5–10] for delivery and encapsulation of biomolecules by leveraging intra- and supramolecular order/disorder[11–20]. Unlike thermodynamically stable compartments, these dynamic assemblies[7,21–26], can be designed to respond to specific stimuli[10,12,25,27,28], and they allow for control of various properties, including polarity, rheology, and surface tension. Yet, the exact molecular mechanisms underlying the formation of biomolecular condensates have not been fully elucidated, although a number of advances have been recently made[29–31]. Thus, the design of biomolecular condensates, or liquid droplets, with tunable chemical and physical functionalities 'on demand' remains a challenge.

[1]Shmunis School of Biomedicine and Cancer Research, George S. Wise Faculty of Life Sciences, Tel Aviv University, Tel Aviv 69978, Israel. [2]Department of Pure and Applied Chemistry, Technology and Innovation Centre, University of Strathclyde, 99 George Street, Glasgow G1 1RD, UK. [3]Department of Chemistry, Faculty of Exact Sciences, Bar Ilan University, Ramat Gan 52900, Israel. [4]Center for Nanoscience and Nanotechnology Tel Aviv University, Tel Aviv 69978, Israel. [5]Sagol Center for Regenerative Biotechnology Tel Aviv University, Tel Aviv 69978, Israel. [6]Center for the Physics and Chemistry of Living Systems Tel Aviv University, Tel Aviv 69978, Israel, Tel Aviv 69978, Israel. ✉e-mail: Jordan.Chill@biu.ac.il; ayalalampel@tauex.tau.ac.il

Experimental molecular-level studies of protein LLPS are performed using recombinant IDPs, which suffer from several limitations. In particular, IDPs have undetermined structure and their preparation involves multistep expression in living cells and purification, which in some cases produce limited yields and require stringent storage conditions. Compared with protein production, peptide synthesis is straightforward and does not require complex expression/purification steps, yet IDPs undergo LLPS at lower, nano- or micromolar concentrations while peptides typically have higher critical LLPS concentrations. Importantly, unlike proteins, changes to composition of peptides, even at the single-amino acid level, directly dictates the supramolecular structure and material properties[32–35], thereby enabling to establish sequence-structure and structure-function relationships.

To gain insights into the driving forces of biomolecular condensates formation, here we use systematic sequence variants of a designer peptide as minimalistic building blocks of synthetic condensates. We design a library of LLPS-promoting peptides that self-coacervate into liquid droplets with tunable chemical and material properties. Since the analytical methods traditionally used to characterize self-assembled peptides such as X-ray scattering techniques are limited for solid-like assemblies[36], we use complementary methodologies including fluorescence recovery after photobleaching (FRAP), Raman spectroscopy and nuclear magnetic resonance (NMR) spectroscopy to shed light on the mechanism of droplet formation, both at the material- and the molecular-level. Our findings show that the peptide sequence controls the LLPS propensity and the material properties of the resulting droplets including mobility and diffusion, as well as the encapsulation efficiency of fluorescent payloads. Moreover, these findings show that arginine (Arg) interactions with the side chains of aromatic amino acids play a key role in LLPS.

## Results

### Peptide sequence controls LLPS propensity

We sought to design a library of LLPS-promoting peptide building blocks which form synthetic biomolecular condensates with tunable chemical composition and physical properties. We hypothesized that minimalistic variants of protein low complexity domains (LCDs) will self-assemble into liquid droplets through LLPS. Specifically, we envisioned that in order to keep the sequence length relatively short, the peptide composition should include high content of aromatic and basic amino acids that can interact through π–π or cation–π interactions. To test this hypothesis, we designed a primary sequence that contains various LLPS-promoting motifs. Inspired by LCDs of ribonucleoproteins (RNPs) and IDPs that are rich in arginine-glycine (RG) dyad or RGG triad repeats[37], we incorporated three repeats of an RG dyad (Fig. 1a), where glycine (Gly) provides flexibility and arginine (Arg) promotes electrostatic interactions with the terminal carboxylic group, cation- π or π interactions with aromatic amino acids (Fig. 1a). While both lysine (Lys) and Arg have basic side chain groups, the guanidine group of Arg delocalizes the charge due to the π bonded system and thus can promote more versatile binding modes compared to Lys, through cation–π and π–π interactions[38]. Thus, considering that the Arg side chain can interact with side chains of aromatic amino acids, we incorporated tryptophan (Trp) and tyrosine (Tyr) into the peptide sequence at a 1:1 stoichiometry with Arg, with aromatic amino acids positioned at both ends of the sequence to enhance π–π stacking interactions[21]. Finally, we considered the elastin-like polypeptide (ELP) repeating motif VPGXG. This pentapeptide sequence is a common LLPS-promoting motif used in engineered ordered/disordered polypeptides, where X can be any amino acid except proline (Pro), and a hydrophobic amino acid at this position promotes coacervation[39,40]. Moreover, a previous work showed that substituting Val at the first position with Trp induced coacervation of the 15-mer ELP (WPGVG)3[41]. Building on these previous findings, we incorporated the sequence WPGVG, thus obtaining the 14-mer peptide WGRGRGRGWPGVGY termed WGR-1 (Figs. 1a, 2a). WGR-1 forms droplets by self-coacervation at neutral pH in the presence of 0.2 M NaCl, which reduces the electrostatic repulsion of the basic peptide (Fig. 1b). We created a phase diagram of WGR-1 as a function of pH and peptide concentration in tris buffer by gradually increasing the pH and monitoring LLPS, as indicated by appearance of sample turbidity (Fig. 2b), where all experiments performed at room temperature. As expected, increasing peptide concentration decreased the pH in which visible turbidity and droplets are observed, where the critical LLPS concentration is 8 mM. To confirm that the turbidity is a result of LLPS and droplet formation rather than aggregation, we used bright field laser scanning confocal microscopy (Fig. 2c).

To shed light on the role of each domain in LLPS, we designed five additional sequence variants (Fig. 2a). Analyzing the LLPS propensity of each sequence variant showed that omitting the Tyr at the C-terminal position (WGR-2) completely arrests LLPS, as no droplets were formed at peptide concentration 5–30 mM in the pH range of 3–12 (Fig. 2b). Substituting Tyr with phenylalanine (Phe) (WGR-3) recovers droplet formation. To study the role of the ELP domain in droplet formation, we omitted this motif from the peptide sequence (WGR-4). To our surprise, removing the ELP domain does not inhibit LLPS, but instead shifts the boundaries of the phase diagram, with higher pH value at the critical LLPS concentration of 8 mM (pH = 9.5 for WGR-4 vs. 8.5 for WGR-1). We attribute this to the higher charge density in WGR-4, with consequent stronger repulsion, when compared to the other LLPS-forming peptides. Reducing the number of RG dyads from three to two (WGR-5) shifts the phase diagram boundaries with droplets formed at 5 mM (Fig. 2b), suggesting that removal of the basic Arg decreases the electrostatic repulsion between the peptide molecules and as a result promotes intermolecular interactions and droplets formation. By using solution NMR analysis (see "Experimental" Section), we found that the pKa of the N-terminal amine group is in the 7.3–7.5 range (Supplementary Table 2), considerably lower than the expected value in the 8.0–8.5 range. These results might explain the changes to LLPS observed from the phase diagrams, where neutralization of the terminal amine leads to reduced electrostatic repulsion between the peptide molecules, and in turn to LLPS. Strikingly, substituting all three Arg with Lys completely arrested LLPS with no turbidity or droplets observed, albeit some aggregates at low abundancy (Fig. 2b, c). While both Lys and Arg can participate in cation-π interactions, only Arg can form π–π interactions due to the sp$^2$ nitrogen atoms. These results suggest that π-interactions between the guanidium group of Arg side chain and the aromatic amino acid side chains are critical for LLPS (Fig. 1c).

Since this analysis is highly sensitive to pH fluctuation, we also created phase diagrams for all peptides using three different buffers that are optimized for specific pH range: citrate buffer for pH 3–7, tris buffer for pH 7–9 and ammonium bicarbonate for pH 9–12 (Supplementary Fig. 1). While the trend in these phase diagrams is similar to that obtained only in tris buffer, the critical pH for LLPS is lower in citrate buffer for all LLPS-promoting peptides (Supplementary Fig. 1). In addition, a slight difference is observed between WGR-1 and WGR-3 at 10 mM, where LLPS is observed at higher pH for WGR-3 (Supplementary Fig. 1), suggesting that Tyr has a stronger contribution to the intermolecular interactions which mediate LLPS than Phe. Indeed, Wang et. al showed that Tyr–Arg interactions are more significant for phase separation than Tyr–Lys interactions and even more than Phe–Arg[42]. Moreover, the critical LLPS concentration of WGR-4 is lower in ammonium bicarbonate than that in tris buffer (5 mM vs. 8 mM, respectively). Thus, these results suggest that citrate and ammonium bicarbonate promote LLPS. To shed light on this, we performed turbidity assay of WGR-1 at 10 mM using the same conditions used in the phase diagram analysis. Higher turbidity is observed at pH 7 with citrate and at pH 9–10 with ammonium bicarbonate compared to tris

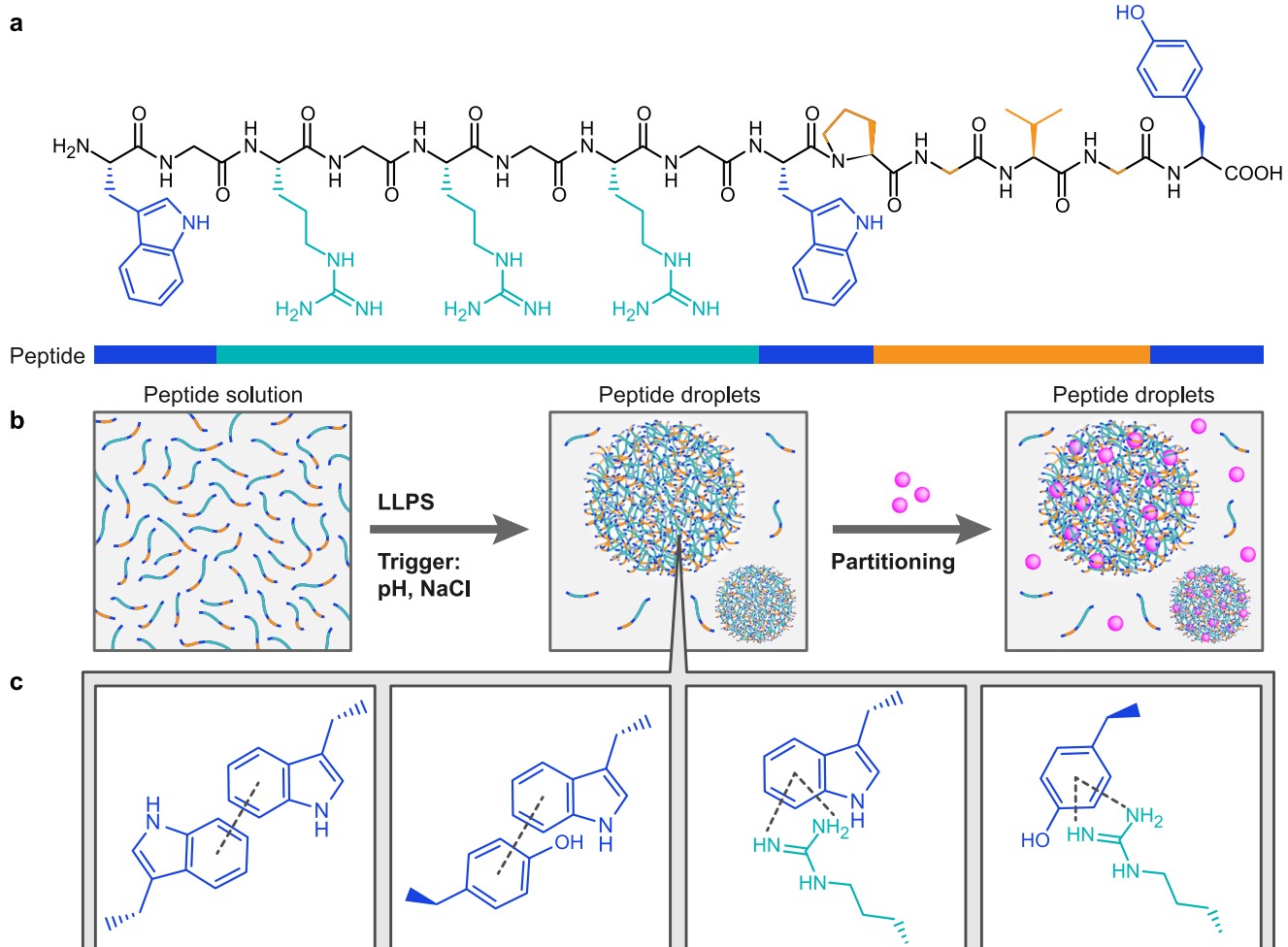

**Fig. 1 | Designer minimalistic peptide droplets. a** Chemical structure of WGR-1. Aromatic amino acid side chains (Trp and Tyr) are colored in blue, Arg side chain is colored in turquoise, and non-polar amino acid side chains that are part of the ELP domain (Pro-Gly-Val-Gly) are colored in orange. **b** Suggested mechanism of the peptide liquid droplet formation and subsequent partitioning of fluorescent payloads. **c** Expected intermolecular interactions underlying LLPS of WGR-1 into liquid droplets, including (from left to right) Trp-Trp, Trp-Tyr, Arg-Trp, and Arg-Tyr π-π stacking. Chemical structures of side chains are presented, color coded as described in (**a**).

buffer (Supplementary Fig. 2), confirming that citrate and ammonium bicarbonate induce peptide LLPS, presumably by reducing the electrostatic repulsion between the peptide molecules provided by their charge state at the respective pH range (−2/−3 for citrate and −1/−2 for ammonium bicarbonate). In contrast, the charge state of tris (+1/0 at the respective pH range) is not expected to reduce this repulsion.

Following these observations, we sought to systematically analyze the effect of ions from the Hofmeister series on LLPS in our minimalistic system. For this, we performed turbidity analysis of WGR-1 at 10 mM in the presence of four different salts that are composed of chaotropic and kosmotropic anions and cations: NaCl, KCl, Na$_2$HPO$_4$, and K$_2$HPO$_4$. We measured sample turbidity at salt concentrations between 10 mM and 200 mM and at three different pH values (6, 7, and 8). When HPO$_4^{2-}$ is used as an anion, sample turbidity appears at pH 8 and no difference in turbidity is observed between K$^+$ and Na$^+$ at concentrations up to 100 mM (Supplementary Fig. 3), which is expected as Hofmeister cations have typically a smaller effect on LLPS and K$^+$ and Na$^+$ are adjacent in the series. At 200 mM, lower sample turbidity is observed for K$_2$HPO$_4$ compared with Na$_2$HPO$_4$. This result correlates with previous reports on the stabilizing effect of K$^+$ on proteins at high mM concentrations[43]. Interestingly, with Cl$^-$ as the anion, Na$^+$ induces LLPS at pH 8 while K$^+$ does not, further showing the stabilizing effect of K$^+$ on the peptide[43]. Moreover, at pH 6, high

turbidity is observed for both KCl and NaCl at low salt concentrations (10 and 50 mM). In a recent work, Knowles and co-workers proposed that LLPS occurs at low pH is mediated by electrostatic interactions, while that occurs at basic pH is mediated by hydrophobic interactions through a salting-out process[44]. Similarly, our results suggest that at pH 6, low Cl$^-$ concentration promotes LLPS by reducing the repulsion between the basic peptide groups, while at pH 8, high concentration of NaCl (but not KCl) induces salting-out of the peptide molecules, where the latter undergo LLPS through various modes of interactions, including π−interactions.

Next, we studied whether the peptide sequence affects the material properties of the resulting liquid droplets. For this, we performed FRAP analysis using laser scanning confocal microscopy of 0.5% FITC-labeled peptides. At this concentration, the dye has a negligible effect on peptide LLPS (Supplementary Fig. 4). The apparent diffusion coefficient for each peptide was calculated as

$$D_{app} = \frac{r^2}{t} \qquad (1)$$

where $t$ is the recovery time. The calculated apparent diffusion coefficients of the peptides were in the $1.7–5.5*10^{-14}$ m$^2$ s$^{-1}$ range (Supplementary Table 1). Similar values were reported for condensates that

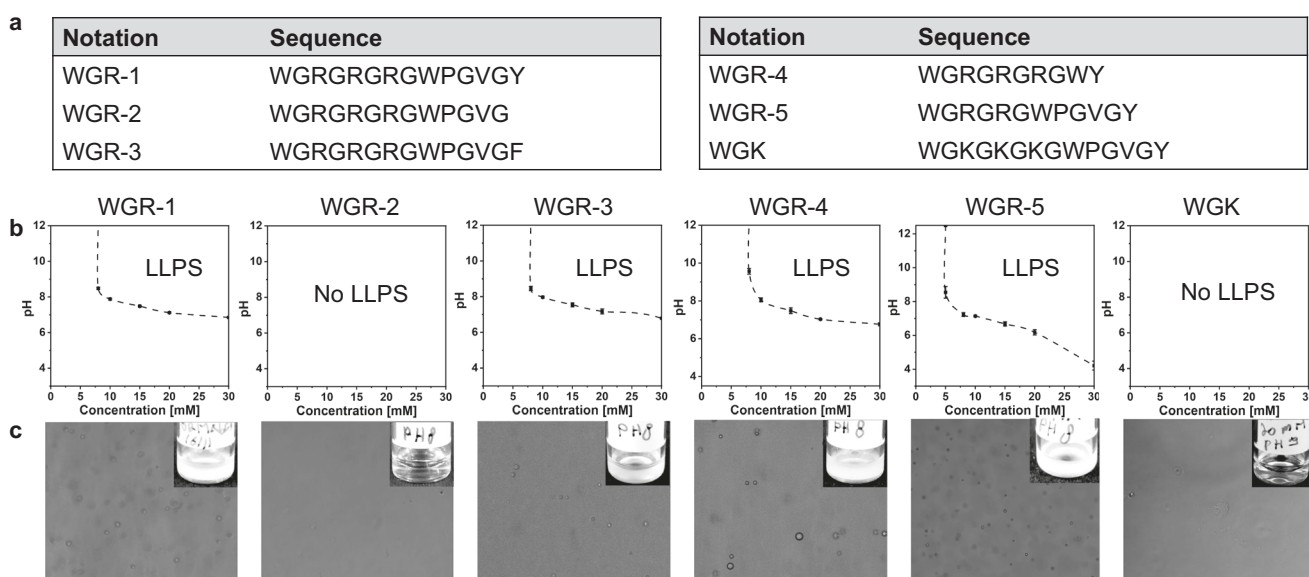

**Fig. 2 | Peptide sequence controls LLPS propensity and droplet formation.**
**a** Table of the designed peptide sequences. **b** Phase diagram of the peptides as a function of peptide concentration and pH, in tris buffer with 0.2 M NaCl, at room temperature. LLPS was not observed for WGR-2 and WGK. Data are presented as mean values of $n = 3$ +/− SD. Source data are provided as a Source Data file.

**c** Confocal microscopy bright field images of peptide liquid droplets formed at a concentration of 20 mM in Tris buffer at pH 8 with 0.2 M NaCl. Scale bars = 20 µm. Inset: macroscopic images of the peptide solutions. 'Source data are provided as a Source Data file.

are formed by complex coacervation of cationic peptide polymers and nucleic acids[38]

Out of the four LLPS-promoting peptides, WGR-3 (substitution of Tyr with Phe) has the largest apparent diffusion coefficient ($D$), more than 5-fold larger than that of WGR-5 and slightly larger than that of WGR-1 (Fig. 3 and Supplementary Table. 1). The higher $D$ of WGR-3 compared to WGR-1 correlates with the LLPS propensity of the two peptides (Supplementary Fig. 1), suggesting that the higher mobility of WGR-3 is a result of weaker interactions between Phe and Arg compared to those of Tyr and Arg[42,45]. WGR-4 has a lower diffusion coefficient than WGR-1, suggesting that the ELP domain interferes sterically with the interactions between the aromatic amino acid side chains, or between the aromatics and Arg, and thus, removing this domain might increase the accessibility of the aromatics and Arg groups. Since the ELP sequence increases the flexibility of the overall peptide, we hypothesized that it could facilitate intra-molecular interactions, with the presence of the bend-promoting Pro residue further enhancing this tendency. Such intra-peptide contacts—if formed—might compete with inter-peptide contacts necessary for LLPS and in turn affect droplet dynamics. By using solution NMR, we followed changes in $^{13}C\alpha$ chemical shifts at low non-LLPS concentrations (3 mM) in which inter-peptide contacts are less likely to occur, upon addition of 8 M urea, expected to perturb intra-molecular contacts. Urea-induced $^{13}C\alpha$ shifts of residues Pro[10] and Val[12] are consistent with an increase in random coil conformation and a decrease in turn conformation in WGR-1, WGR-3, and WGR-5, but not in WGR-2, lacking the aromatic residue required for intra-peptide interactions (Supplementary Table 3). Shifts of other residues (i.e., Trp[1]) do not exhibit this difference. Notably, one of our findings distinguishes WGR-3 from all other peptides as striking urea-induced $^{13}C$ shifts (in the 0.2–0.5 ppm range) for Trp[9] C$\alpha$, C$\beta$ and aromatic C$\gamma$ were observed only in WGR-3 spectra. These findings suggest that the ELP domain induces intramolecular interactions, yet LLPS is obviously influenced by many different factors. The lowest diffusion of WGR-5 indicates that decreasing the electrostatic repulsion by reducing the net charge of the peptide from +3 to +2 increases the strength of intermolecular interactions between the peptide building blocks and as a result, significantly lowers the mobility and

dynamics of the droplets (Fig. 3a–c). Moreover, these results suggest that in the absence of the aliphatic ELP domain, cation–π or π–π interactions between Arg and the aromatic side chain are the dominant driving force for droplet formation. As these interactions are short-range, they can result in higher friction between the peptides molecules, and in turn, reduced peptide diffusion and droplet dynamics[37,38]. Peptide diffusion in the dilute phase, as calculated by solution NMR analysis, is in the $1.9–2.4*10^{-10}$ $m^2$ $s^{-1}$ range (Supplementary Table 1).

## Droplet encapsulation efficiency is influenced by peptide hydrophobicity

Next, we studied how peptide composition affects the encapsulation efficiency of the droplets by using GFP, rhodamine B, and fluorescein as fluorescent payload model systems (Fig. 4). We analyzed the encapsulation efficiency of the fluorescent payloads by using both confocal microscopy and absorbance measurements of the payloads in the dilute vs. the condensed phase (Fig. 4a–c). The encapsulation efficiency (EE) of the fluorescent payloads ranges between 72 and 99% (Fig. 4d), where the hydrophobic Phe-containing peptide WGR-3 has the lowest EE of GFP, and the most polar peptide, WGR-4, has the highest EE of GFP and of rhodamine B, suggesting that the peptide interacts with the dye either electrostatically, by π–π, or cation–π interactions. WGR-1 has the highest EE of fluorescein. In addition to π-π interactions, WGR-1 might form hydrogen bonding with the hydroxyl groups or electrostatic interactions with the deprotonated carboxylic acid of the dye. Thus, these results demonstrate that the encapsulation efficiency of the compartments can be modulated by the chemical composition of the peptide building blocks.

## π-π interactions and hydrogen bonding underlie peptide droplet formation

To gain molecular-level insights into the network of intermolecular interactions which underlie droplet formation, we performed extensive Raman and NMR spectroscopy analyses. We envisioned that employing these complementary techniques, which provide information on molecular interactions of assemblies in the solid-state (Raman spectroscopy) and in solution (NMR), might facilitate a

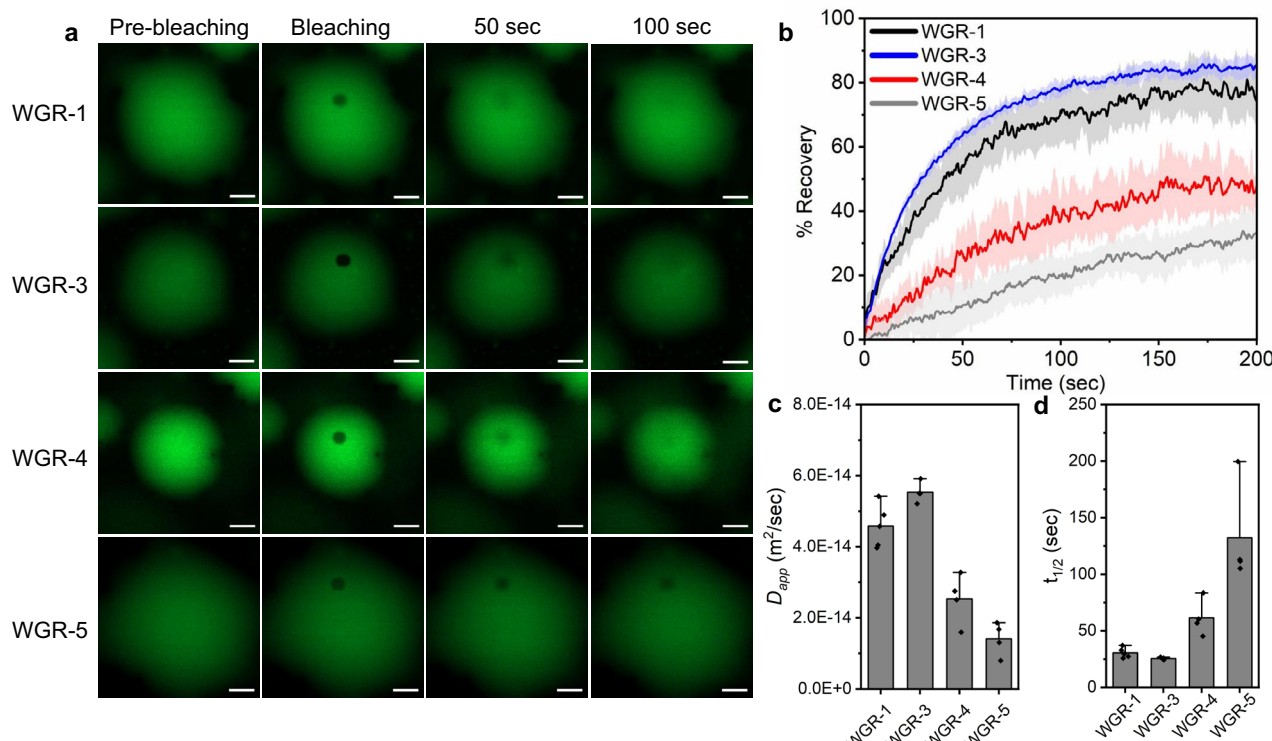

**Fig. 3 | Strength of intermolecular interactions affects peptide droplet dynamics. a–c** FRAP analysis of WGR-1, WGR-3, WGR-4, and WGR-5, performed using laser scanning confocal microscopy at 20 mM in Tris buffer at pH 8 with 0.2 M NaCl using 0.5% FITC-labeled peptides. **a** Representative confocal microscopy images of FRAP for individual droplets. Scale bars = 5 μm. FRAP recovery plots (**b**), apparent diffusion coefficient (**c**), and $t_{1/2}$ of the recovery (**d**). Data are presented as mean values +/− SD, $n$ = 5 (WGR-1), 4 (WGR-3, WGR-4), and 5 (WGR-5). Source data are provided as a Source Data file.

more holistic understanding of the mechanism that governs droplet formation.

To analyze the peptide droplets by Raman spectroscopy, WGR-1 droplets were drop-casted on a precoated glass substrate (see "Experimental" section), and solution Raman spectra of the droplets were collected. An average solution Raman spectrum of WGR-1 droplet sample is shown in Fig. 5a. Upon analysis of the representative spectrum, we found that this and other significant peaks originated from the Trp side chains. Most of the peaks are associated with C−H bending, ring-stretching, and deformation in the indole ring (1551, 1010, 876, and 758 $cm^{-1}$). The peaks at 1433 and 1615 $cm^{-1}$ are attributed to the symmetric and asymmetric stretching of the $COO^-$ group and the 1573 $cm^{-1}$, related to $NH_3^+$ vibrations[46]. These structural markers can be used to assign the interactions which underlie droplet formation. It has been previously shown that the Raman bands at 1551, 1358, and 1010 $cm^{-1}$ are strongest when the indole ring is hydrogen bonded[47]. These three peaks are very prominent in the peptide droplet spectra suggesting that hydrogen bonding is a critical interaction for droplet formation. The 876 $cm^{-1}$ peak is an indole ring vibration mode associated with a displacement of the $N_1H$ group nearly along the $N_1$–H bond which decreases upon hydrogen bonding[46]. In our spectra, this peak is weak, further evidence for the involvement of hydrogen bonding in droplet formation. Finally, slight shifting of the 1010 $cm^{-1}$ band to the 1009–1010 $cm^{-1}$ range suggests a loss of van der Waals interactions within the droplet. We also observe a doublet (850/830) that originates from Tyr side chain. Conflicting explanations of the 850/830 ratio of peaks were previously reported[48,49] and thus its interpretation in our system is not obvious, yet it is clearly sensitive to the hydrophobicity of the phenol environment[49,50].

Next, we sought to analyze individual droplets by using Raman mapping. Droplets in solution cannot be mapped due to their mobility, thus we dried the drop-casted droplets, mapped them, and created

false color 2D and 3D images (Fig. 5b, c) by plotting the intensity of the 758 $cm^{-1}$ peak throughout the imaged area, in the center and edge of the droplet as well as in the surrounding phase. Similar spectra were obtained from dried droplets (Supplementary Fig. 5) compared with solution droplets (Fig. 5a), suggesting that drying the droplet did not significantly alter the interactions that mediate droplet formation. Yet, we did lose some information from the Tyr doublet, which is very weak in the dried spectrum. Within the droplet itself, we see some differences in the Raman spectra throughout the mapped area. Notably, spectral differences in the 700–800 $cm^{-1}$ and 1300–1500 $cm^{-1}$ regions were observed between the center (white) and the edge (red) of the droplet, as shown in the normalized Raman spectra (Fig. 5b, d). The intensity ratio between the 1360/1340 $cm^{-1}$ peaks is higher at the droplet center when compared to its edge, an indicator of increased hydrophobicity within the condensed phase. We attribute the weaker spectrum at the droplet edge in the 1300–1400 $cm^{-1}$ region to focusing. No peptide signal is observed in the spectrum of the surrounding phase. The main difference in the 700–800 region is the shift in the 758 $cm^{-1}$ peak from the droplet center, to 765 $cm^{-1}$ at the edge. A similar shift in the relative intensity of 759 $cm^{-1}$ was previously attributed to cation−π interactions of the model compound, diaza crown ether with two indole substituents[51]. Thus, the shift observed between the center and the edge of the droplet indicates the involvement of Trp in cation−π interactions.

To further confirm the role of Trp in LLPS, we designed and studied 4 additional sequence variants, where we omitted Trp at position 1 (WGR-6), omitted Trp at position 9 (WGR-7), substituted Trp at position 9 with Ala (WGR-8), and both omitted Trp at position 1 and substituted Trp at position 9 with Ala (WGR-9). None of the peptides undergoes LLPS at all conditions tested (concentrations up to 30 mM and at pH 3–11). Optical microscopy analysis of the peptides at 30 mM shows clear solutions and some amorphous aggregates

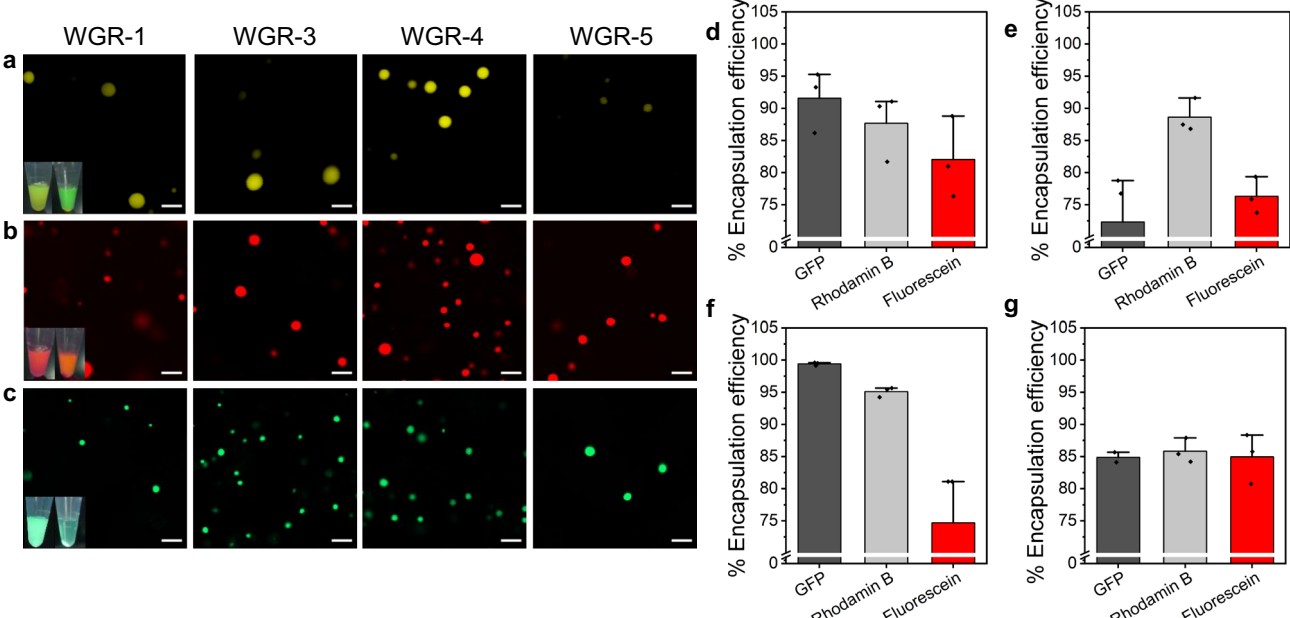

**Fig. 4 | Partitioning of fluorescent payloads within droplets depends on peptide polarity.** Confocal microscopy images of (**a**) fluorescein, (**b**) Rhodamine B and (**c**) GFP partitioning within WGR-1, WGR-3, WGR-4, and WGR-5 peptide droplets. Scale bar = 10 μm. Insets: Macroscopic images of the partitioning of the fluorescent payload in peptide droplets samples before (left) and after (right) centrifugation and droplet sedimentation. Encapsulation efficiency (EE) analysis of peptide droplets calculated from absorbance measurements of bulk solutions for (**d**) WGR-1 (**e**) WGR-3 (**f**) WGR-4 (**g**) WGR-5. Data are presented as mean of $n = 3$ +/− SD. Source data are provided as a Source Data file.

(Supplementary Fig. 6). These results strengthen the findings from the Raman spectroscopy and highlight the critical role of Trp side chains in LLPS.

## NMR points to the molecular mechanism of peptide droplet formation

For a solution view of this system, we employed NMR, well-known for its unique ability to provide information on the conformations of low-complexity disordered peptides and their motions on a wide range of timescales[52–55]. A combination of homonuclear 2D-$^1$H,$^1$H-COSY/TOCSY and heteronuclear 2D-$^1$H,$^{13}$C-HMQC/HMBC spectra acquired for 20 mM WGR-1 at pH 6 without NaCl (non-LLPS conditions) afforded an assignment of $^1$H/$^{13}$C chemical shifts of this peptide. Using a similar array of experiments, the vast majority of these assignments could then be transferred and reassigned in peptides at various pH values and NaCl concentrations. Observable chemical shifts for monomeric WGR-1 under all conditions (see Supplementary Tables 4, 5 and Supplementary Fig. 9) were consistent with a disordered peptide in random coil conformation, in agreement with the results from the CD analysis (Supplementary Fig. 7). Towards the peptide C-terminus side chains exhibited a double (major/minor) set of shifts with a typical intensity ratio of ~2.2:1, attributed to *trans/cis* isomers of the P$^{10}$ pyrrolidine ring.

Although slowly tumbling peptides in the condensed phase are typically inaccessible to high-resolution NMR, equilibrium between the two phases results in chemical shift changes for the bulk phase, reporting on molecular changes induced by droplet formation and identifying intermolecular interactions contributing to this process. In doing so we focused on the two known LLPS-inducing factors for this system, salinity and pH. Since such changes are inherently small, we based our analysis of shifts under LLPS-promoting conditions on a comparison to non-LLPS-promoting conditions.

$^{13}$C resonance frequencies were followed for 5 and 20 mM WGR-1 samples at pH 8 with increasing NaCl concentrations (Supplementary Data 1). Whereas the 5 mM sample remains translucent throughout the titration (non-LLPS conditions), the 20 mM sample exhibits

coacervation at higher concentrations (LLPS conditions). Thus, the difference in NaCl-induced spectral changes between the two samples (arbitrarily chosen at 0 and 0.1 M) is an indication of the effects of LLPS (Fig. 5e). The six Gly (lacking side chains) were omitted from this analysis, and due to spectral overlap resonances of the three Arg side chains were grouped together. While side chain-specific NaCl-induced changes were in the 1.4–2.5 Hz range for the 5 mM sample (with the exception of W$^1$, due to slight changes in the pKa of the terminal NH$_3^+$ group), they were significantly larger, in the 5–7 Hz range, for the 20 mM sample. The differential change averaged 3.5 Hz and was relatively uniform throughout the WGR-1 sequence, with the largest difference observed for residue P$^{10}$ (-5.5 Hz). Since these changes reflect the indirect effect of droplet environment upon the NMR-visible bulk peptide, they confirm the Raman results regarding the LLPS-induced environmental change and suggest a global effect upon the peptide.

We then turned to examine the effects of pH increase as an inducer of LLPS. As expected, higher pH-induced changes in chemical shifts clustered around ionizable groups of the peptide at the amino terminus (pKa ~ 7.5, Supplementary Table 2) and the Tyr$^{14}$ phenolic ring (pKa ~ 10). In contrast, smaller yet still significant differences observed between shifts at pH 10 and concentrations of 5 (non-LLPS conditions) and 20 mM (LLPS conditions, Fig. 6a, b and Supplementary Fig. 8) were instructive in pointing to molecular changes accompanying droplet formation. The most significant differences (between 5 and 20 mM samples at pH 10) observed in the $^{13}$C NMR spectrum (>0.1 ppm) were for the aromatic ring $^{13}$C nuclei of Tyr$^{14}$ side chain (Fig. 6a). Specifically, the changes seen for the γ- and ε-$^{13}$C (but not the δ-$^{13}$C) suggest an electron-donating effect of the anionic phenol group, a hypothesis consistent with the tangible change in the ζ-$^{13}$C resonance (Fig. 6a). These findings implicate the Tyr$^{14}$ aromatic ring as a key determinant of coacervation. Raman-observed changes involving the H$^{ε1}$–N$^{ε1}$ bond in the Trp indole rings are undetectable by NMR under basic conditions, and generally relatively small chemical shifts were observed for the Trp carbons.

Reasoning that an intermolecular interaction is necessary for coacervation to become favorable, a likely candidate for this

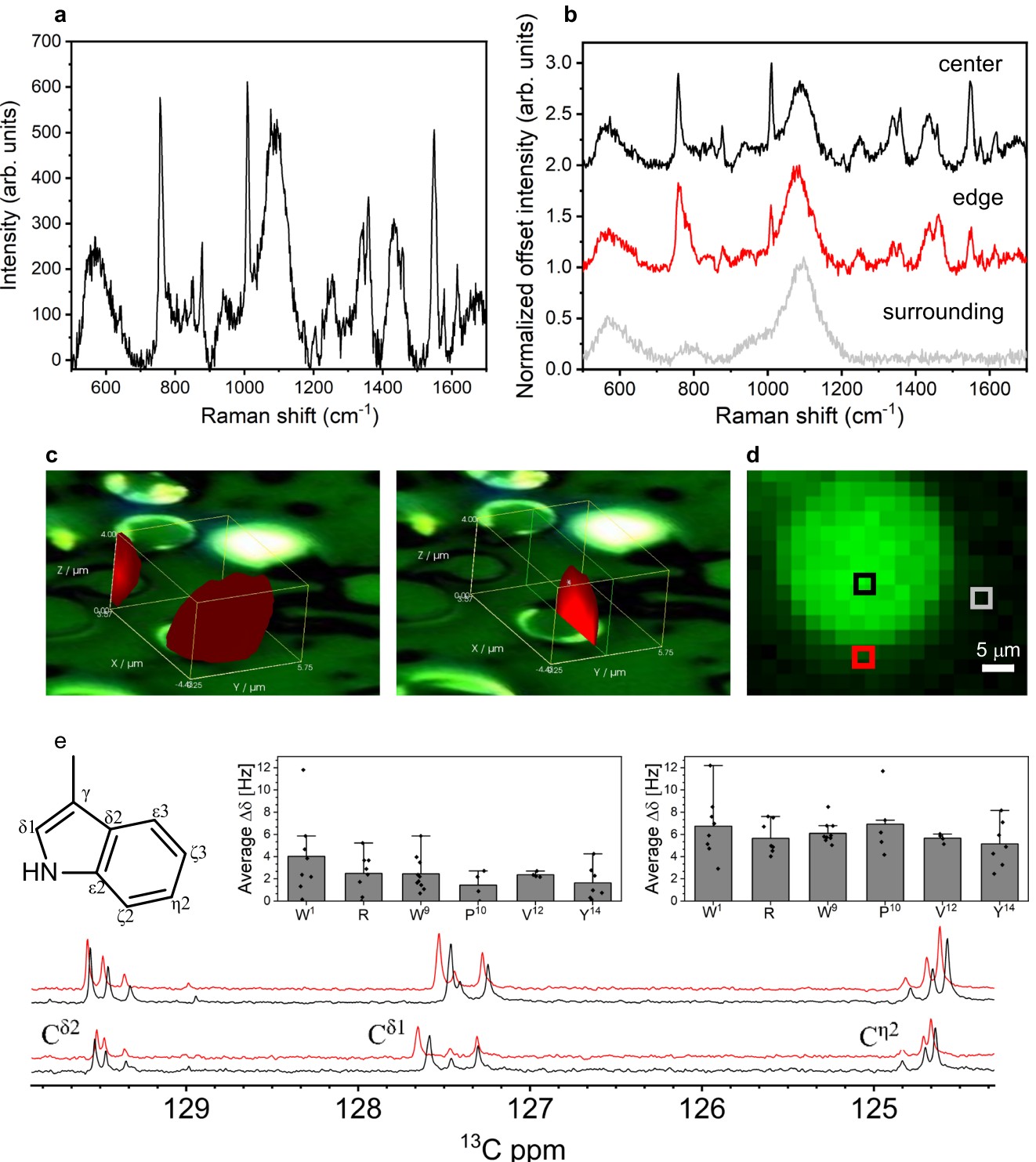

**Fig. 5 | Molecular level analysis of droplet formation by using Raman and NMR spectroscopy. a** Raman spectrum obtained from averaging solution Raman map. **b** Normalized Raman spectra taken from 3 different spots of the 2D false color image: droplet center (black), droplet edge (red) and the droplet surrounding (light grey). The large peak at 1100 cm⁻¹ originates from the glass background. **c** False color 3D image showing whole peptide droplet and slice through the center. **d** False color 2D image created using the 758 cm⁻¹ peak. . One-dimensional $^{13}$C spectrum of WGR-1 peptide with (red) or without (black) 100 mM NaCl, at

5 mM peptide (non-LLPS, lower spectra) and 20 mM peptide (upper spectra) in 50 mM tris buffer pH 8 and 300 K, chemical shifts are assigned. Inset: averaged $\Delta\delta$ in $^{13}$C spectra of the four samples for each amino acid at peptide concentration of 5 mM (left bar chart) and 20 mM (right bar chart). Gly were excluded from the analysis due to spectral overlap, and all three Arg were averaged due to partial spectral overlap. Data are presented as mean values +/− SD, $n = 8$ (W1), 8 (R), 11 (20 mM W9), 10 (5 mM W9), 5 (20 mM P), 4 (5 mM P), 4 (V), 7 (Y). Source data are provided as a Source Data file.

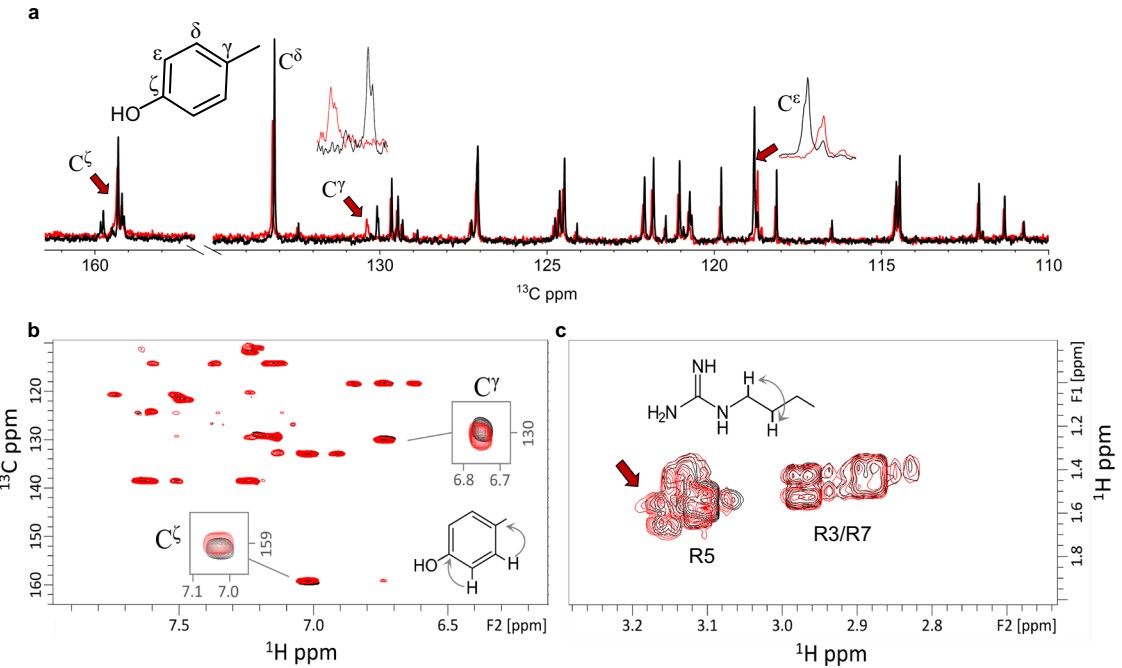

**Fig. 6 | NMR determines the molecular mechanism of droplet formation. a** The one-dimensional $^{13}$C spectrum for the WGR-1 peptide at 20 (black, LLPS) and 5 (red, non-LLPS) mM in 50 mM tris buffer pH 10 and 300 K. Arrows indicate chemical shift differences at the $Y^{14}$ aromatic ring. **b** Aromatic region of the 2D-$^1$H,$^{13}$C-HMBC spectrum showing long-range proton-carbon correlations allowing the detection of quaternary carbons. $Y^{14}$ chemical shift changes are shown as before. **c** Region of the 2D-$^1$H, $^1$H-COSY spectrum showing the correlation between arginine $H^\gamma$–$H^\delta$ protons.

intermolecular contact is an Arg sidechain whose positively charged π-system is known for its ability to interact with electron-rich aromatic rings, such as Tyr[14]. This would explain the loss of LLPS in the WGK peptide (Fig. 2b), in which all three Arg at position 3, 5, and 7 were replaced by Lys that are similarly positively charged yet unsuitable for π−π stacking interactions. We assumed that such an interaction must cause chemical shift perturbations in a second site along the peptide, and to this end focused upon chemical shifts of Arg $H^\delta$ nuclei, located closest to the guanidino π-electron system and best detected using their COSY cross-peak with the neighboring $H^\gamma$ protons located in a distinguishable spectral region. While Arg[3] and Arg[5] cross-peaks were mostly unaffected by the concentration increase from 5 to 20 mM, the third—representing Arg[7] as determined from our assignment—exhibited a concentration-induced change (Fig. 6c and Supplementary Fig. 10). We conclude that the key intermolecular contact points in the formation of droplets for WGR-1 are the sidechains of Arg[7] and Tyr[14], with effects upon all residues in this segment. A similar pattern of Arg[7] chemical shift changes was observed for WGR-3, in which Tyr[14] is replaced by Phe[14] (Supplementary Fig. 11). Thus, the two complementary techniques allowed the detection of different intermolecular interactions that drive droplet formation, where the solid-state analysis suggests the involvement of Trp hydrogen bonding and π−π interactions, and the solution-state analysis indicates the role of Tyr/Phe and Arg π-interactions.

## Discussion

We have developed a library of LLPS-promoting peptide building blocks that form synthetic biomolecular condensates with varying chemical composition and biophysical properties. Our findings show that the peptide chemical composition directly affect LLPS propensity and droplet formation, even at a single amino acid level. We show that the material properties of the droplets can be tuned by changes to the peptide sequence, where electrostatic repulsion, steric hindrance, and specific intra- and intermolecular interactions directly affects peptide diffusion. Specifically, our findings suggest that intramolecular contacts between Tyr/Phe and Arg side, induced by the ELP domain, might compete with intermolecular interactions in the condensed droplet phase, resulting in accelerated diffusivity. In turn, these sequence changes can be applied to tune the encapsulation efficiency of designed biomolecular condensates. To gain molecular level understanding of the peptide-peptide interactions that underly droplet formation, we combined solid-state analysis of droplets by Raman spectroscopy and NMR solution-state analysis. We found that Trp side chains participate in intermolecular interactions within the droplet center (Raman spectroscopy), and that the interaction between Tyr/Phe and Arg is crucial for droplet formation (NMR analysis). The latter finding emphasizes previous evidence of the critical role of Arg (rather than Lys) interactions with aromatic side chains in formation of cellular[42] and lab-based[38] biomolecular condensates. To summarize, this work demonstrates that minimalistic designer peptides are attractive building blocks for biomolecular condensates with tuneable material properties. This approach opens tremendous opportunities to further develop tuneable and customizable peptide biomolecular condensates as delivery and microreactor systems.

## Methods

### Materials

Peptides were custom synthesized, then purified by high performance liquid chromatography to 95% and supplied as lyophilized powders by Genscript, Hong Kong. Unless otherwise specified, all reagents were of the highest available purity. Fluorescein, rhodamine B and ammonium bicarbonate were purchased from Holland Moran. NaCl, NaOH and HCl were purchased from BioLab, Trizma base was purchased from Sigma. Citrate and citric acid were purchased from Tzamal. GFP (Abcam) was purchased from Zotal as a solution of 1 mg/ml in 0.316% Tris HCl, 10% glycerol at pH 8 that was aliquoted and stored at −20 °C until use.

### Phase diagrams

150 μl of 5 mM, 8 mM, 10 mM, 15 mM, 20 mM, and 30 mM peptide solutions were prepared in either 20 mM tris buffer or 20 mM of the

following three buffers: citrate buffer for pH 3–7 tris buffer for pH 7–9 and ammonium bicarbonate for pH 9–12, with 0.2 M NaCl. The pH was increased gradually until a turbidity appeared and measured as described below. All measurements were performed at room temperature. Data points represent averages of three independent measurements. Turbidity of 35 µl samples was estimated in triplicates from sample absorbance at λ = 500 nm as described below.

## Turbidity measurements

150 µl of 20 mM peptide solutions were prepared in 20 mM Tris buffer. The pH was adjusted to the desired value of 6, 7, 8, 9, 10, and 11 then the turbidity of 35 µl was measured in triplicates at λ = 500 nm using a BioTek H1 synergy plate reader (purchased from Lumitron, Israel).

## Secondary structure evaluation by Circular Dichroism (CD)

Samples solutions for CD were prepared at concentration of 1 mM in 20 mM Tris buffer solution with and without 0.2 M NaCl and were placed in a 0.1 mm path length quartz cuvette at 25 °C, and the range of 190–260 nm was recorded on a Chirascan spectrometer. Background (buffer with or without NaCl according to the sample) was subtracted from the CD spectra.

## Imaging

All samples were imaged in a 96-well Black Glass bottom plate, glass 1.5H (produced by Hangzhou Xinyou, and purchased from Danyel Biotech) 10 min after preparation. The images were taken by Zeiss Zen 900 confocal microscope with ×20/0.8 NA Plan- Apochromat air objective. Images were collected and processed using Zen software (Zeiss). The light microscopy images were taken at PMT mode. PMT imaging were taken with 561 nm laser and fluorescence imaging were taken with 488, 561, and 405 nm lasers for fluorescein, rhodamine B and GFP, respectively.

## Fluorescence recovery after photobleaching

FRAP experiments were performed by a Zeiss Zen 900 confocal microscope with ×20/0.8 NA Plan- Apochromat air objective. For each of the peptides, out of a total peptide concentration of 20 mM we used 0.5% FITC-labeled peptide, in tris buffer pH 8 with 0.2 M NaCl. A circular area with radius of 2.5 µm was bleached with a 488 nm laser 100% intensity at 10 iterations; subsequent recovery of the bleached area was recorded with a 488 nm laser. Monitoring of fluorescence recovery in the condensates was analyzed using Zen Blue 3.2 software (Zeiss). Photobleaching correction and recovery time were calculated using OriginLab 9.95. The final FRAP recovery curve is the average of recovery curves collected from $n = 4$–6 separate droplets.

## Encapsulation efficiency

Stock solutions (1 mM) of the fluorescein and rhodamine B dye molecules were prepared in 20 mM Tris buffer. Coacervate solutions of 20 mM peptide were prepared in 20 mM Tris + 0.2 M NaCl at pH 8. From these, a volume of 148.5 µl was mixed with 1.5 µl of dye solution in a 1.5 ml Eppendorf tube and pipetted. After 10 min the samples were centrifuged at 15,000 × $g$ for 10 min. A volume of 120 µl from the supernatant was collected and vortexed and then the absorbance of 35 µl triplicates was measured (at λ = 490 nm for fluorescein and λ = 555 nm for RhB) in a 384 well black plate by Biotek H1 synergy plate reader (purchased from Lumitron, Israel). For GFP, a 7.5 µl of 20 mM Tris buffer to a 7.5 µl GFP aliquot from the purchased stock solution. The 15 µl of GFP solution was added to 135 µl of peptide solution and was mixed in a 1.5 ml Eppendorf tube and pipetted. The concentration of GFP at the supernatant was measured via fluorescence. All experiments were performed in triplicate. The concentration of the supernatant solutions determinate by calibration curves. Imaging was made to 30 µl of uncentrifuged samples. Efficiency of encapsulation

(%EE) was calculated using Eq. 2.

$$\%EE = \frac{C_T - C_{sup}}{C_T} \qquad (2)$$

## Raman spectroscopy

Samples of 10 mM WGR-1 in 20 mM Tris + 0.2 M NaCl peptide droplets were prepared at pH 8 and analyzed using Raman spectroscopy in solution and when dried. To analyze the droplets in solution, 5 µL of the solution was placed onto a glass slide which was precoated with sigmacote. Average Raman spectra was obtained by focusing on the surface of the liquid droplet using a 20× magnification objective lens and Raman mapping an area with a step size of 20 µm, using a 532 nm laser excitation with 8 mW laser power and a 10 s integration time. To analyze a dried droplet, 1 µL of the solution was drop-casted onto an uncoated glass slide. Excess buffer was removed by washing the dried droplet with water and again left to dry. The droplet was focused on using a 100× magnification objective and Raman mapped with a step size of 0.5 µm, using a 532 nm laser excitation with a laser power of 8 mW. All data was collected and analyzed using WiRE 4.2 software. MATLAB_R2019b was used for map processing.

## NMR

NMR samples were prepared by dissolving peptides in *ca.* 500 µl of 20 mM Tris buffer and 5% $^2H_2O$ supplemented with the appropriate concentration (0–0.2 M) of NaCl and adjusted to the desired pH using dilute HCl or NaOH. Measurements were conducted on a DRX700 Avance-III Bruker spectrometer using a cryogenic triple-resonance TCI or RT-TXI probe-head equipped with $z$-axis pulsed field gradients. All spectra were acquired at a field of 16.4 (700.45 and 176.12 MHz for $^1H$ and $^{13}C$ nuclei, respectively) and at 300 K unless otherwise indicated. Chemical shift assignment was performed using data from one- and two-dimensional NMR experiments run with standard Bruker library files and acquisition parameters, including 1D $^1H$, 1D-$^{13}C$, 2D-$^1H$,$^1H$-COSY, 2D-$^1H$,$^1H$-TOCSY, 2D-$^1H$,$^1H$ ROESY, 2D-$^{13}C$,$^1H$ HMQC (set to $^1J$ correlations) and 2D-$^{13}C$,$^1H$ HMBC (set to $^{2,3}J$ correlations) experiments. Typical mixing times for TOCSY and ROESY experiments were 150 and 200 ms, respectively. Spectra were processed and visualized using the Bruker TopSpin 3.6 software suite. To obtain differential shift changes, following the assignment of peaks for 5 and 20 mM WGR-1 at 0 and 100 mM NaCl (a total of four 1D-$^{13}C$ spectra) $\Delta\delta$ values were calculated using the equation $\Delta\delta = \delta_{100} - \delta_0$, where $\delta_{100}$ and $\delta_0$ are the chemical shifts at 100 and 0 mM NaCl, respectively. The average $\Delta\delta$ was then calculated by averaging $\Delta\delta$ values for all non-overlapping $^{13}C$ nuclei in each amino acid. Gly residues were eliminated from the analysis due to overlaps in the spectrum, and the three Arg residues were pooled together due to partial overlaps in the spectrum. The W[1] α, β, and γ carbons were excluded from the analysis due to the local NaCl-induced effect upon the pKa of the N-terminal $NH_3^+$. For amino acids exhibiting two peaks due to the P10 cis-trans equilibrium the aggregate $\Delta\delta$ was a 0.7:0.3 weighted average of values measured for the major and minor peaks. Uncertainties were determined by the spectral resolution and adjusted to account for ambiguous assignments in the case of Trp[1]/Trp[9].

## Statistics & reproducibility

No statistical method was used to predetermine sample size. No data were excluded from the analyses. The experiments were not randomized. The investigators were not blinded to allocation during experiments and outcome assessment. Microscopy images represent at least three independent analyses.

## Reporting summary

Further information on research design is available in the Nature Portfolio Reporting Summary linked to this article.

## Data availability

All data generated or analysed during this study are included in this published article (and its supplementary information files). Source data are provided with this paper.

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

## Acknowledgements

This research was supported by the Israel Science Foundation, Grant No. 2589/21 (A.L.). We thank Dr. M. Afri for help with NMR experiments and analysis. T.M. thanks the ADAMA Center for Novel Delivery Systems in Crop Protection for the PhD ADAMA Fellowship and the Marian Gertner Institute for Medical Nano systems for the Gertner scholarship.

## Author contributions

A.B.L. and A.L. conceived and designed the experiments. Peptide LLPS experiments and analysis were conducted by A.B.L. and S.B.D. Raman experiments and analyses were performed by S.S.D., K.F., and D.G. NMR experiments were conducted by H.G. and A.B.L. NMR analyses were conducted by J.C., H.G., and T.M. A.L., A.B.L., and J.C. wrote and edited the manuscript. All authors discussed and commented on the manuscript.

## Competing interests

The authors declare no competing interests.
