## [Peer Review File · Nature Communications]

REVIEWER COMMENTS

Reviewer #1 (Remarks to the Author):

The manuscript presents an investigation of LLPS of a series of short peptides. The series is designed to highlight the contributions of pi-cation interactions on the LLPS formation and their properties. This reductionist approach is powerful, however the manuscript can benefit from additional quantitative analysis. For example, how many pi-cation interactions are needed in order to archive LLPS? How strong are pi-cation compared to pi-pi? How strong are they compared to charge-charge interactions? How much the LLPS will be affected in these cases by salt concentrations? Such basic questions can strengthen the manuscript particularly that its motivation is in basic principles of LLPS.

Comments:

1. I wonder on the molecular effect of the ELP region on LLPS. Particularly, what is the origin of the difference between WGR1 and WGR4 that show LLPS but at different conditions and a shift of the critical point toward a basic regime with lower protein concentration?
2. Understanding the difference between intra and inter-chains of WGR1-5 is essential to understand their different LLPS behaviour. Is it possible to speculate on the ratio of intra and inter cation-pi between Arg and F/W/Y?
3. How different are the diffusion of these proteins in the dense phases with respect to a synthetic polyampholyte condensate phase dominated by electrostatic interactions only? Could you comment on the ratio of diffusivity of proteins within droplet and in dilute solution for the designed peptides?
4. In order to correlate the enhanced interactions coupled slowed-down diffusivity it is valuable to quantify how strong is Arg-Y interactions compared to Arg-F interactions that leads to almost 5 times higher diffusivity in the dense phase of WGR-3 than 5?

Reviewer #2 (Remarks to the Author):

Leshem et al. present an interesting peptide undergoing simple coacervation. They show changes in the sequence of the peptide capable to tune material properties and recruitment of small molecules. Fundamental sequence-function studies are highly required in the field, and this peptide may prove very useful for practical applications in material sciences. However, several clarifications and major revisions seem required. A key advantage of these interesting peptides is the possibility to link sequence (and encoded interactions) to the observed macroscopic behavior in details. This possibility is not yet fully exploited. See in particular points 2 and 4 below.

- 1) Figure 2: solution conditions should be specified also in the main text, as well as temperature. With the resolution of the data the difference between WGR-1 and WGR-3 seems negligible. WGR-1 sample was analyzed at pH 8 and WGR-3 at pH 7. More points seem required, or alternatively this part should be removed. The two peptides are also similar in the FRAP analysis (Fig. 3). This is important since later the authors discuss the importance of Y14. These data show that F14 leads to very similar behavior. In general, the data acquired to build the phase diagrams are very scarce. More points should be taken for all peptides. Error bars and evidence of reproducibility must also be reported.

2) The authors should clearly explain the reason of the change of C_{sat} at pH values lower than 6 for WGR-1, as well as the pH dependence of the other peptides. For instance, WGR-5 has a drastic change at pH 7. There are no histidines in the sequence. The authors point later to Arg-Tyr interactions. Are these interactions (or others) modulated by changes of pH in the range 6-7? Since citrate buffer is used at pH 6 and tris at pH 7, the authors should check for specific ion effects. E.g. carefully comparing citrate and tris buffer in the pH range 6-7. NMR analysis at lower pH and/or additional experiments at different salt concentrations and salt types (e.g. Hofmeister series) and/or at different temperatures can provide evidence and further support the proposed fundamental interactions underlying the observed pH dependence.

3) Figure 3: solution conditions and peptide concentration should be reported in the caption and/or in the main text. Peptides were 100% labelled with FITC? Does FITC labelling affect the C_{sat} of the peptides? Half-times could be plotted against measured C_{sat} under the same conditions to highlight the expected correlation between LLPS and material properties (the deeper inside the phase diagram the higher the viscosity).

4) I have serious concerns about the Raman analysis due to the drying of the sample. Desolvation drastically changes the interactions. There are examples where interactions in the dense phase were resolved by NMR. The proposed role of Trp should be at least verified by mutational analysis similar to the ones performed for Y/F and R/K.

5) For IDPs in some cases intramolecular interactions leading to chain compaction correlate with intermolecular interactions responsible for LLPS. The authors could comment on this aspect for their peptides.

6) Although not necessary, it would add value to the work the characterization of the concentration of peptides in the dense phase, or the water content.

Other points:

7) Introduction: "a process termed complex coacervation": in some cases, biological LCDs promote simple coacervation (as in this work).

8) "that is mainly entropy-driven": this can be misleading. Several biological LCDs exhibit UCST behavior and LLPS is therefore enthalpy-driven.

9) "IDPs have complex sequence and structure": this is unclear. IDPs lack determined structures. Moreover, sequence is not more complex than any standard protein.

10) "limited yields": this can largely depend on the IDP. In this comparison, it could be noted that IDPs undergo LLPS in the nano-micromolar range, while peptides (as in this work) require mM concentrations.

11) Conclusions: "Our findings show that the peptide chemical composition... markedly affect LLPS propensity" This is rather known

12) "Material properties can be tuned over a wide range": the reported data do not seem to be in a wide range

13) Experimental section, turbidity measurements: 150 ul of 20 mM?

14) Experimental section: FITC labelling is not reported.

Point by point responses to the reviewers' comments:

Reviewer #1

The manuscript presents an investigation of LLPS of a series of short peptides. The series is designed to highlight the contributions of pi-cation interactions on the LLPS formation and their properties. This reductionist approach is powerful, however the manuscript can benefit from additional quantitative analysis. For example, how many pi-cation interactions are needed in order to achieve LLPS? How strong are pi-cation compared to pi-pi? How strong are they compared to charge-charge interactions? How much the LLPS will be affected in these cases by salt concentrations? Such basic questions can strengthen the manuscript particularly that its motivation is in basic principles of LLPS.

Response: We thank the reviewer for the positive feedback. Please see our responses for each of the comments below.

1. I wonder on the molecular effect of the ELP region on LLPS. Particularly, what is the origin of the difference between WGR1 and WGR4 that show LLPS but at different conditions and a shift of the critical point toward a basic regime with lower protein concentration?

Response: We thank the reviewer for the constructive comment. Regarding the WGR-1/WGR-4 difference in terms of a shift to a more basic regime, we attribute this to the fact that in WGR-1 the electrostatic repulsion effect is somewhat tempered by the addition of the contact-promoting ELP sequence. Thus, LLPS can form in WGR-1 even before any ionization of the Tyr¹⁴ phenol group occurs, whereas in WGR-4 significant (~20%) ionization must occur for LLPS to form.

However, this should not be considered the 'role' of the ELP sequence. Since the ELP sequence increases the flexibility of the overall peptide, we hypothesized that it could facilitate intra-molecular interactions (i.e., between Arg⁷ and Tyr¹⁴ of *the same* WGR-1), with the presence of the bend-promoting proline residue further enhancing this tendency. Such intra-peptide contacts – if formed – might compete with inter-peptide contacts necessary for LLPS (assuming equality in all other factors). We followed changes in ¹³Ca

chemical shifts (good indicators of backbone conformation) at low non-LLPS concentrations (3 mM, at which inter-peptide contacts are less likely to occur) upon addition of 8 M urea (expected to perturb peptide-peptide contacts). As we report in the text, urea-induced ^{13}C shifts of residues Pro¹⁰ and Val¹² are consistent with an increase in random coil conformation and a decrease in turn conformation in WGR-1, WGR-3, and WGR-5, but not in WGR-2 lacking the aromatic residue required for intra-peptide interactions (WGR-4 was not included in this analysis since it lacks the ELP sequence). Shifts of other residues (i.e., Trp¹) do not exhibit this difference. Thus, these findings suggest that the ELP domain induces intramolecular interactions.

We included the solution NMR urea analysis to Supplementary Table 3 (NMR SI section) and added the following text to the revised manuscript (page 9 line 2): “Since the ELP sequence increases the flexibility of the overall peptide, we hypothesized that it could facilitate intra-molecular interactions, with the presence of the bend-promoting Pro residue further enhancing this tendency. Such intra-peptide contacts – if formed – might compete with inter-peptide contacts necessary for LLPS and in turn affect droplet dynamics. By using solution NMR, we followed changes in ^{13}C chemical shifts at low non-LLPS concentrations (3 mM) in which inter-peptide contacts are less likely to occur, upon addition of 8 M urea, expected to perturb intra-molecular contacts. Urea-induced ^{13}C shifts of residues Pro¹⁰ and Val¹² are consistent with an increase in random coil conformation and a decrease in turn conformation in WGR-1, WGR-3, and WGR-5, but not in WGR-2, lacking the aromatic residue required for intra-peptide interactions (Supplementary Table 3). Shifts of other residues (i.e., Trp¹) do not exhibit this difference. Notably, one of our findings distinguishes WGR-3 from all other peptides as striking urea-induced ^{13}C shifts (in the 0.2-0.5 ppm range) for Trp⁹ C α , C β and aromatic C γ were observed only in WGR-3 spectra. These findings suggest that the ELP domain induces intramolecular interactions, yet LLPS is obviously influenced by many different factors.”

Page 6 line 4: “To our surprise, removing the ELP domain does not inhibit LLPS, but instead shifts the boundaries of the phase diagram, with higher pH value at the critical LLPS concentration of 8 mM (pH=9.5 for WGR-4 vs. 8.5 for WGR-1). We attribute this to the higher charge density in WGR-4, with consequent stronger repulsion, when compared to the other LLPS-forming peptides.”

2. Understanding the difference between intra and inter-chains of WGR 1-5 is essential to understand their different LLPS behaviour. Is it possible to speculate on the ratio of intra and inter cation-pi between Arg and F/W/Y?

Response: We thank the reviewer for the question. As detailed above (in response to comment 1), our solution NMR analysis suggest that in all peptides bearing the ELP domain there is a contribution of intramolecular contacts. Notably, one of our findings distinguishes WGR-3 (bearing Phe instead of Tyr) from all other peptides. In our urea experiment (performed at low peptide concentration), striking urea-induced ^{13}C shifts (in the 0.2-0.5 ppm range) for Trp⁹ C α , C β and aromatic C γ were observed only in WGR-3 spectra; significantly, Trp¹ exhibited normal urea-induced shifts. We consider these shifts to report on intramolecular interactions. Yet, it is difficult to speculate on the ratio of intra/inter-molecular interactions as the magnitude of the urea effect described above is small and would correspond to ~5-10% of intra-peptide interactions. Therefore, it would

require several more peptides and measurements with complementary approaches to confirm this estimate.

The results from the urea experiments are presented in Supplementary Table 3 (NMR SI section) and the text was revised accordingly (as detailed in response to comment 1).

3. How different are the diffusion of these proteins in the dense phases with respect to a synthetic polyampholyte condensate phase dominated by electrostatic interactions only? Could you comment on the ratio of diffusivity of proteins within droplet and in dilute solution for the designed peptides?

Response: The reported diffusivity of condensates formed by complex coacervation of cationic peptide polymers and nucleic acids is similar to the one reported here, with $D=2.18 \times 10^{-13} \text{ m}^2 \text{ sec}^{-1}$ for poly-K and $2.9 \times 10^{-15} \text{ m}^2 \text{ sec}^{-1}$ for poly-R (Fisher and Elbaum-Garfinkle, *Nat. Commun.*, 2019)

To clarify this point, we added the following text to the revised manuscript (page 8 line 15): “Similar values were reported for condensates that are formed by complex coacervation of cationic peptide polymers and nucleic acids³⁸.”

To address the comment in full, we have measured the diffusion of the peptides in the dilute phase using solution NMR. As expected, the diffusion of the peptides in the dilute phase was found to be 4-orders of magnitude faster than that in the condensed phase, with values of $D=1.91\text{E-}10$, $2.18\text{E-}10$, $2.23\text{E-}10$ and $2.44\text{E-}10 \text{ m}^2 \text{ sec}^{-1}$ for WGR-1, WGR-3, WGR-4, and WGR-5, respectively.

We included these results in revised Supplementary Table 1 and added the following text to the manuscript (page 9 line 22): “Peptide diffusion in the dilute phase, as calculated by solution NMR analysis, is in the $1.9\text{-}2.4 \times 10^{-10} \text{ m}^2 \text{ sec}^{-1}$ range (Supplementary Table 1).”

4. In order to correlate the enhanced interactions coupled slowed-down diffusivity it is valuable to quantify how strong is Arg-Y interactions compared to Arg-F interactions that leads to almost 5 times higher diffusivity in the dense phase of WGR-3 than 5?

Response: Our detailed NMR analysis did not find a significant difference between the Arg⁷-Tyr¹⁴ and Arg⁷-Phe¹⁴ interactions. Yet, as detailed above in responses to comments 1 and 2, we did find that only WGR-3 has urea-induced ¹³C shifts for Trp⁹ C α , C β and aromatic C γ , suggesting that this peptide forms additional intramolecular contacts that might decrease the strength of intermolecular interactions and in turn increase the D . We attribute the slowed-down diffusivity of WGR-5 to the reduced electrostatic repulsion of this peptide and as a result, increased peptide-peptide interactions.

The new NMR analyses are presented in Supplementary Table 3, and Supplementary Fig. 11.

Reviewer #2

Leshem et al. present an interesting peptide undergoing simple coacervation. They show changes in the sequence of the peptide capable to tune material properties and recruitment of small molecules. Fundamental sequence-function studies are highly required in the field, and this peptide may prove very useful for practical applications in material sciences. However, several clarifications and major revisions seem required. A

key advantage of these interesting peptides is the possibility to link sequence (and encoded interactions) to the observed macroscopic behavior in details. This possibility is not yet fully exploited. See in particular points 2 and 4 below.

1. Figure 2: solution conditions should be specified also in the main text, as well as temperature. With the resolution of the data the difference between WGR-1 and WGR-3 seems negligible. WGR-1 sample was analyzed at pH 8 and WGR-3 at pH 7. More points seem required, or alternatively this part should be removed. The two peptides are also similar in the FRAP analysis (Fig. 3). This is important since later the authors discuss the importance of Y14. These data show that F14 leads to very similar behavior. In general, the data acquired to build the phase diagrams are very scarce. More points should be taken for all peptides. Error bars and evidence of reproducibility must also be reported.

Response: We thank the reviewer for this important comment. We acknowledge that the experimental protocol of the phase diagrams in its original version might not be clear enough, and would like to clarify this. First, we performed all experiments at room temperature and added this information to the main text. To obtain phase diagrams for each of the peptides, we originally screened a range of peptide concentrations: 5 mM, 10 mM, 20 mM and 30 mM. For each of these concentrations, we gradually increased the pH of the samples until turbidity appeared (citrate buffer was used for pH range of 3-7, tris buffer for pH range of 7-9 and ammonium bicarbonate for pH range of 9-12). Thus, each of the data points in the phase diagrams indicates the pH in which turbidity appears, as an indication of LLPS.

To clarify our experimental protocols for the phase diagrams, we revised the manuscript text as follows:

Page 5 line 16: “We created a phase diagram of WGR-1 as a function of pH and peptide concentration in tris buffer by gradually increasing the pH and monitoring LLPS, as indicated by appearance of sample turbidity (Fig. 2b), where all experiments performed at room temperature.”

Figure 2 caption: “b. Phase diagram of the peptides as a function of peptide concentration and pH, in tris buffer with 0.2 M NaCl, at room temperature. Values represent average pH of three independent measurements, error bars are indicated.”

To address the reviewer’s comment, we added data points to the phase diagrams (found in revised Figure 2b and Supplementary Fig. 1). Specifically, we added two concentrations to the phase diagrams: 8 mM and 15 mM and added error bars to all data points, which represent the standard deviation of the critical pH for LLPS of three independent measurements (from three independent samples). Moreover, to address the next comment made by the reviewer regarding the possible effect of the different ions used in the phase diagrams (in citrate/tris/ammonium bicarbonate buffers), we created new phase diagrams for all peptides only in tris buffer. The results and conclusions from this analysis are detailed below in response to comment 2. Notably, we found that the critical LLPS concentration of WGR-1 and WGR-3 is 8 mM, and the critical pH for LLPS of WGR-4 at 5 mM in ammonium bicarbonate buffer is 8.1, which is not within the range of the optimal pH for this buffer. We discuss these effects in the revised manuscript and compare the phase diagrams created using different buffers/in tris (more details below).

We added the new results to Figure 2b and Supplementary Fig. 1, and revised the text accordingly (page 5 line 19): “As expected, increasing peptide concentration decreased the pH in which visible turbidity and droplets are observed, where the critical LLPS concentration is 8 mM.”

Experimental Section: “**Phase diagrams: 150 μ l of 5 mM, 8 mM, 10 mM, 15 mM, 20 mM and 30 mM peptide solutions were prepared in either 20 mM tris buffer or 20 mM of the following three buffers:** citrate buffer for pH 3-7, tris buffer for pH 7-9 and ammonium bicarbonate for pH 9-12, with 0.2 M NaCl. The pH was increased gradually until a turbidity appeared and measured as described below. **All measurements were performed at room temperature. Data points represent averages of three independent measurements.** Turbidity of 35 μ l samples was estimated in triplicates from sample absorbance at $\lambda=500$ nm as described below.”

Considering the reviewer’s comments about the difference between WGR-1/WGR-3, the reviewer correctly mentions the similar LLPS propensity and dynamics of the two peptides. Our new data, and specifically the new phase diagrams performed in tris buffer, confirms this similarity (Figure 2b). To gain insight on the interactions that underlie LLPS when Phe replaces Tyr, we repeated the solution NMR analysis reported in the manuscript for WGR-1, this time applying it to WGR-3 and found a similar changes in Arg⁷ as those observed for WGR-1, suggesting that Arg⁷ interacts with Phe¹⁴.

The new NMR analysis are presented in Supplementary Fig. 11. The following text was added to the revised manuscript (page 14 line 23): “**A similar pattern of Arg⁷ chemical shift changes was observed for WGR-3, in which Tyr¹⁴ is replaced by Phe¹⁴ (Supplementary Fig. 11).**”

2. The authors should clearly explain the reason of the change of Csat at pH values lower than 6 for WGR-1, as well as the pH dependence of the other peptides. For instance, WGR-5 has a drastic change at pH 7. There are no histidines in the sequence. The authors point later to Arg-Tyr interactions. Are these interactions (or others) modulated by changes of pH in the range 6-7? Since citrate buffer is used at pH 6 and tris at pH 7, the authors should check for specific ion effects. E.g. carefully comparing citrate and tris buffer in the pH range 6-7. NMR analysis at lower pH and/or additional experiments at different salt concentrations and salt types (e.g. Hofmeister series) and/or at different temperatures can provide evidence and further support the proposed fundamental interactions underlying the observed pH dependence.

Response: We thank the reviewer for the important comment - after exhaustive analyses performed to address this comment in full, we indeed found that the different salts, and especially citrate, have different effects on peptide LLPS. In the revised manuscript, we discuss these effects and draw conclusions on the role of peptide sequence in LLPS, by taking into account ions effect and carefully comparing sequences using the same ions, as detailed below.

We have performed the following extensive analyses:

- i) Analyzed the effect of the different buffers used in this work on LLPS: citrate, tris, and ammonium bicarbonate buffer;
- ii) Analyzed the effect of different salts from the Hofmeister series on LLPS;

iii) Performed solution NMR analysis to determine the pKa of the terminal amine for each of the peptides.

The results of each analysis are detailed below:

i) To analyze the effect of the ions in each of the buffers on peptide LLPS, we performed turbidity assay of 10 mM WGR-1 using citrate buffer at pH 6-7, tris buffer at pH 6-10 and ammonium carbonate buffer at pH 9-10, with 200 mM NaCl. For citrate buffer, we measured sample turbidity at pH 6, the optimal value for this buffer, and in addition at pH 7, to compare the effect of citrate and tris on LLPS. Similarly, we analyzed the turbidity in ammonium bicarbonate buffer at pH 9 in addition to pH 10, to compare the effect of tris and ammonium bicarbonate on LLPS.

Higher turbidity is observed at pH 7 with citrate compared to tris and with ammonium bicarbonate at pH 9-10 compared to tris. These results suggest that citrate and ammonium bicarbonate facilitate LLPS, presumably by reducing the electrostatic repulsion between the peptide molecules provided by their charge state at the respective pH range (-2/-3 for citrate and -1/-2 for ammonium bicarbonate). In contrast, the charge state of tris (+1/0 at the respective pH range) is not expected to reduce this repulsion, and therefore has an 'inert effect' on LLPS.

Following these results and in order to draw direct links between peptide sequence and LLPS, excluding ions effects, we performed an additional analysis and *created new phase diagrams for all peptides in tris buffer*. While the phase diagrams in tris are similar to those obtained with citrate/tris/ammonium bicarbonate buffers, it is clear that citrate buffer lowers the critical pH for all LLPS-promoting peptides. In addition, a slight difference is observed between WGR-1 and WGR-3 at 10 mM, where LLPS is observed at higher pH for WGR-3 (Supplementary Fig. 1), suggesting that the phenol side chain group of Tyr participates in intermolecular interactions which mediate LLPS. Indeed, Wang et. al showed that Tyr-Arg interactions are more significant for phase separation than Tyr-Lys interactions and even more than Phe-Arg¹. Moreover, ammonium bicarbonate lowers the critical LLPS concentration of WGR-4 (5 mM in ammonium bicarbonate compared with 8 mM in tris).

In light of these findings and the obvious ion effect, we decided to present the new phase diagrams in tris in main text **Figure 2b** and the phase diagrams with the different buffers in **Supplementary Fig. 1**. The text was revised accordingly (see revised manuscript, revised text is highlighted in red). In addition, the turbidity assay of the different buffers was added to **Supplementary Fig. 2**. The following text was added to the revised manuscript

Page 6 line 21: "Since this analysis is highly sensitive to pH fluctuation, we also created phase diagrams for all peptides using three different buffers that are optimized for specific pH range: citrate buffer for pH 3-6, tris buffer for pH 7-9 and ammonium bicarbonate for pH 9-12 (Supplementary Fig. 1). While these phase diagrams are similar to those obtained only in tris buffer, the critical pH for LLPS is lower in citrate buffer for all LLPS-promoting peptides (Supplementary Fig. 1). In addition, a slight difference is observed between WGR-1 and WGR-3 at 10 mM, where LLPS is observed at higher pH for WGR-3 (Supplementary Fig. 1), suggesting that the phenol side chain group of Tyr participates in intermolecular interactions which mediate LLPS. Indeed, Wang et. al showed that Tyr-Arg interactions are more significant for phase separation than Tyr-Lys interactions and even more than Phe-

Arg¹. Moreover, the critical LLPS concentration of WGR-4 is lower in ammonium bicarbonate than that in tris buffer (5 mM vs. 8 mM, respectively). Thus, these results suggest that citrate and ammonium bicarbonate promote LLPS. To shed light on this, we performed turbidity assay of WGR-1 at 10 mM using the same conditions used in the phase diagram analysis. Higher turbidity is observed at pH 7 with citrate and at pH 9-10 with ammonium bicarbonate compared to tris buffer (Supplementary Fig. 2), confirming that citrate and ammonium bicarbonate induce peptide LLPS, presumably by reducing the electrostatic repulsion between the peptide molecules provided by their charge state at the respective pH range (-2/-3 for citrate and -1/-2 for ammonium bicarbonate). In contrast, the charge state of tris (+1/0 at the respective pH range) is not expected to reduce this repulsion.”

ii) To gain insights on how **ions from the Hofmeister series affect LLPS** in our minimalistic system, we performed turbidity analysis of WGR-1 at 10 mM in the presence of four different salts that are composed of chaotropes and kosmotropes anions and cations: NaCl, KCl, Na₂HPO₄ and K₂HPO₄. We measured sample turbidity at salt concentrations between 10 mM-200 mM and at three different pH (6, 7 and 8). When HPO₄²⁻ is used as an anion, sample turbidity appears at pH 8 and no difference in turbidity is observed between K⁺ and Na⁺ at concentrations up to 100 mM. This result is expected as Hofmeister cations typically have a smaller effect on LLPS and K⁺ and Na⁺ are adjacent in the series. At 200 mM, lower sample turbidity is observed for K₂HPO₄ compared with Na₂HPO₄. This result correlates with previous reports on the stabilizing effect of K⁺ on proteins at high mM concentrations. Interestingly, when Cl⁻ used as anion, Na⁺ induces LLPS at pH 8 while K⁺ does not, further confirming the stabilizing effect of K⁺. Moreover, at pH 6, high turbidity is observed for both KCl and NaCl, only at lower salt concentrations of 10 mM and 50 mM. In a recent work, Knowles and co-workers proposed that LLPS occurs at low pH is mediated by electrostatic interactions, while that occurs at high pH is mediated by hydrophobic interactions through a salting-out process² (Kraimer *Nat. Commun.* 2021). Based on this proposition, our results might suggest that at pH 6, low KCl and NaCl concentration promote peptide LLPS, mediated in part by electrostatic interactions, where Cl⁻ ions reduce the repulsion between the basic peptide groups. At pH 8, high NaCl concentration, but not KCl, might induce salting-out of the peptide molecules, where the latter undergo LLPS through various modes of interactions, including π -interactions.

) We agree with the reviewer that LLPS in low pH range (6-7) is indeed not obvious considering that there are no ionizable groups with an expected pKa in this range (i.e., histidine). Following this comment, we sought to study the possible involvement of the N-terminal amine group in LLPS-inducing interactions. We estimated the effective pKa value in our system for the amino terminal group by following Trp¹ chemical shifts (i.e. the conveniently detected H α , H β shifts) along a pH titration in the 6.0-9.0 range. We found that the pKa value of the terminal amino group is in the 7.3-7.5 range, considerably lower than the expected value in the 8.0-8.5 range. We attribute this to the proximity of other positively charged groups (particularly Arg³). This accounts for the changes observed in this pH range – below the pKa, the positively charged amino terminal increases the electrostatic repulsion between peptides and lowers the LLPS tendency. It also accounts for the different result seen for WGR-5, since this peptide carries one less positive charge. The text has been changed accordingly to reflect these insights.

The solution NMR analysis of the pKa for all peptides is included in the NMR section of the revised Supplementary Information as Supplementary Table 2.

The following text was included in the revised manuscript (page 6 line 11): “By using solution NMR analysis (see Experimental Section), we found that the pKa of the N-terminal amine group is in the 7.3-7.5 range (Supplementary Table 2), considerably lower than the expected value in the 8.0-8.5 range. These results might explain the changes to LLPS observed from the phase diagrams, where neutralization of the terminal amine leads to reduced electrostatic repulsion between the peptide molecules, and in turn to LLPS.”

3. Figure 3: solution conditions and peptide concentration should be reported in the caption and/or in the main text. Peptides were 100% labelled with FITC? Does FITC labelling affect the Csat of the peptides? Half-times could be plotted against measured Csat under the same conditions to highlight the expected correlation between LLPS and material properties (the deeper inside the phase diagram the higher the viscosity).

Response: We thank the reviewer for pointing out the missing information. For the FRAP analysis, we used a final peptide concentration of 20 mM in tris buffer at pH 8 in the presence of 0.2 M NaCl. To minimize the effect of the hydrophobic and aromatic FITC dye on peptide-peptide interactions and LLPS, we used only 0.5% of FITC-labeled peptide out of the final 20 mM concentration.

Yet, to address the comment in full, we analyzed the effect of FITC-labeling on WGR-1 LLPS using turbidity assay. The results show that at a peptide concentration of 10 mM, the presence of 0.5% FITC-labeled peptide increases sample turbidity. However, at a peptide concentration of 20 mM, the turbidity of WGR-1 samples containing 0.5% FITC-labeled peptide is slightly lower, suggesting that the effect of FITC at this concentration is negligible.

We added the turbidity analysis to Supplementary Figure 4, and added the following text to the revised manuscript (page 8 line 9): “For this, we performed fluorescence recovery after photobleaching (FRAP) analysis using laser scanning confocal microscopy of 0.5% FITC-labeled peptides. At this concentration, the dye has a negligible effect on peptide LLPS (Supplementary Fig. 4).”

In addition, we revised Figure 3 caption accordingly: “...FRAP analysis of WGR-1, WGR-3, WGR-4, and WGR-5, performed using laser scanning confocal microscopy at 20 mM in tris buffer at pH 8 with 0.2 M NaCl using 0.5% FITC-labeled peptides....”

The following text was added to the Experimental Section: “**Fluorescence recovery after photobleaching:**For each of the peptides, out of a total peptide concentration of 20 mM we used 0.5% FITC-labeled peptide, in tris buffer pH 8 with 0.2 M NaCl.”

4. a) I have serious concerns about the Raman analysis due to the drying of the sample. Desolvation drastically changes the interactions. There are examples where interactions in the dense phase were resolved by NMR.

(b) The proposed role of Trp should be at least verified by mutational analysis similar to the ones performed for Y/F and R/K.

Response: We thank the reviewer for both comments. Our responses below:

a) Originally, our attempts to obtain Raman spectra of droplets failed due to droplet movement, therefore we obtained spectra of dried droplet samples. To address the reviewer's comment, we used a glass coating protocol, which we already used in our recently published work³ (Katzir, Haimov and Lampel, *Adv. Mater.*, 2022, 2206371). The pre-coated glass assisted in keeping the droplet movement to a minimum, and thus we succeeded in obtaining Raman spectra from the droplets in solution. The obtained spectrum of droplets in solution is similar to the spectrum of dried droplets, suggesting that drying the droplet did not significantly alter the interactions that mediate droplet formation. However, a (850/830) doublet which originates from Tyr is present in the solution spectrum but not in the dry droplet spectrum (signal is very weak in the dry-state). While it is clear that the 850/830 peaks ratio is sensitive to the hydrophobicity of the phenol environment^{4, 5}, previous reports suggest conflicting explanations for the ratio^{5, 6}, therefore it is not obvious to interpret it in our system.

Still, we could not perform Raman mapping of individual droplets in solution, even with the pre-coated glass, due to the droplets' slight movement. *In light of the similarity of the solution-state and the dry-state spectra*, and considering that mapping individual droplets provides important insights on the organization of the droplet, we kept both analyses in the revised manuscript. We included the solution-state spectrum in the revised Figure 5a and moved the dry-state spectrum to the SI alongside the spectrum of the solution background (buffer on glass).

The following text was added to the revised manuscript:

Page 10 line 19: "To analyze the peptide droplets by Raman spectroscopy, WGR-1 droplets were drop-casted on a pre-coated glass substrate (see Experimental section), and solution Raman spectra of the droplets were collected. An average solution Raman spectrum of WGR-1 droplet sample is shown in Fig. 5a."

Page 11 line 9: "We also observe a doublet (850/830) that originates from Tyr side chain. Conflicting explanations of the 850/830 ratio of peaks were previously reported^{5, 6} and thus its interpretation in our system is not obvious, yet it is clearly sensitive to the hydrophobicity of the phenol environment^{4, 5}. Next, we sought to analyze individual droplets by using Raman mapping. Droplets in solution cannot be mapped due to their mobility, thus we dried the drop-casted droplets, mapped them, and created false color 2D and 3D images... Similar spectra were obtained from dried droplets (Supplementary Fig. 5) compared with solution droplets (Fig. 5a), suggesting that drying the droplet did not significantly alter the interactions that mediate droplet formation. Yet, we did lose some information from the Tyr doublet, which is very weak in the dried spectrum. Within the droplet itself, we see some differences in the Raman spectra throughout the mapped area...."

) We agree that studying the LLPS propensity of sequence variants to verify the role of Trp can strengthen the work. For this, we designed 4 additional peptides: WGR-6, in which Trp at position 1 was omitted (GRGRGRGWPGVGYa); WGR-7, in which Trp at position 9 was omitted (WGRGRGRGPGVGY), WGR-8, in which Trp at position 9 was substituted with Ala (WGRGRGRGAPVGY), and WGR-9, in which Trp at position 1 was omitted and Trp at position 9 was substituted to Ala (GRGRGRGAPVGY). This analysis shows that none of

the 4 peptides undergoes LLPS at the same conditions used for the phase diagram i.e., concentrations of 5 mM-30 mM at pH range of 3-11.

We added the peptide sequences and the optical microscopy analysis showing no LLPS for all peptides to Supplementary Fig. 6.

The following text was added to the revised manuscript (page 12, line 6): “To further confirm the role of Trp in LLPS, we designed and studied 4 additional sequence variants, where we omitted Trp at position 1 (WGR-6), omitted Trp at position 9 (WGR-7), substituted Trp at position 9 with Ala (WGR-8), and both omitted Trp at position 1 and substituted Trp at position 9 with Ala (WGR-9). None of the peptides undergoes LLPS at all conditions tested (concentrations up to 30 mM and at pH 3-11). Optical microscopy analysis of the peptides at 30 mM shows clear solutions and some amorphous aggregates (Supplementary Fig. 6). These results strengthen the findings from the Raman spectroscopy and highlight the critical role of Trp side chains in LLPS.”

5. For IDPs in some cases intramolecular interactions leading to chain compaction correlate with intermolecular interactions responsible for LLPS. The authors could comment on this aspect for their peptides.

Response: We thank the reviewer for the comment. Since the ELP sequence increases the flexibility of the overall peptide, we hypothesized that it could facilitate intra-molecular interactions, with the presence of the bend-promoting proline residue further enhancing this tendency. To test this hypothesis, we followed changes in ^{13}C chemical shifts (good indicators of backbone conformation) at low non-LLPS concentrations (3 mM, at which inter-peptide contacts are less likely to occur) upon addition of 8 M urea (expected to perturb peptide-peptide contacts). As we report in the text, urea-induced ^{13}C shifts of residues Pro¹⁰ and Val¹² are consistent with an increase in random coil conformation and a decrease in turn conformation in WGR-1, WGR-3, and WGR-5, but not in WGR-2 lacking the aromatic residue required for intra-peptide interactions (WGR-4 was not included in this analysis since it lacks the ELP sequence). Shifts of other residues (i.e., Trp¹) do not exhibit this difference. Thus, these findings suggest that the ELP domain induces intramolecular interactions.

We included the solution NMR urea analysis to Supplementary Table 3 (NMR SI section) and added the corresponding text to describe our finding (see page 9 in the revised manuscript).

6. Although not necessary, it would add value to the work the characterization of the concentration of peptides in the dense phase, or the water content.

Response: We agree that characterizing the concentration of peptides in the dense phase could add interesting insights on this minimalistic system and thus we plan to continue with this characterization in a follow-up work.

7. Introduction: “a process termed complex coacervation”: in some cases, biological LCDs promote simple coacervation (as in this work).

Response: Indeed, in some cases LCDs promote LLPS through simple coacervation. We removed the line “a process termed complex coacervation” from the introduction.

8. “that is mainly entropy-driven”: this can be misleading. Several biological LCDs exhibit UCST behavior and LLPS is therefore enthalpy-driven.

Response: We agree with the comment and therefore omitted this part from the text. The revised line reads as follows: “The commonly suggested mechanism for the formation of biomolecular condensates is based on LLPS of IDPs and other biomolecules (mainly nucleic acids), a process termed complex coacervation^{3,4}, in which the condensates’ building blocks are highly mobile and exchange rapidly with the surrounding environment.”

9. IDPs have complex sequence and structure”: this is unclear. IDPs lack determined structures. Moreover, sequence is not more complex than any standard protein.

Response: This line refers to the difference in complexity between proteins and peptides, where the first are much larger molecules. We agree that this statement might be too general, therefore revised the text as follow: “In particular, IDPs have **undetermined** structure and their preparation....”

10. “limited yields”: this can largely depend on the IDP. In this comparison, it could be noted that IDPs undergo LLPS in the nano-micromolar range, while peptides (as in this work) require mM concentrations.

Response: To address the comment we revised the text as follows (page 3 line 21):

“In particular, IDPs have **undetermined** structure and their preparation involves multistep expression in living cells and purification, which **in some cases** produce limited yields and require stringent storage conditions. Compared with protein production, peptide synthesis is straightforward and does not require complex expression/purification steps, **yet IDPs undergo LLPS at lower, nano- or micromolar concentrations while peptides typically have higher critical LLPS concentrations. Importantly,** unlike proteins, changes to composition of peptides...”

11. Conclusions: “Our findings show that the peptide chemical composition... markedly affect LLPS propensity” This is rather known

Response: We agree with the comment and revised the text as follows: “Our findings show that the peptide chemical composition directly affect LLPS propensity and droplet formation, even at a single amino acid level.”

12. “Material properties can be tuned over a wide range”: the reported data do not seem to be in a wide range

Response: The text was revised accordingly: “We show that the material properties of the droplets can be tuned by changes to the peptide sequence, where electrostatic repulsion, steric hindrance, and specific intermolecular interactions directly affects peptide diffusion.”

13. Experimental section, turbidity measurements: 150 ul of 20 mM?

Response: Thanks for the comment. The line was revised as follows: “150 μ l of **20 mM** peptide solutions were prepared...”

14. Experimental section: FITC labelling is not reported.

Response: Thank you for drawing our attention to this mistake. The following text was added to the Experimental Section: **“Fluorescence recovery after photobleaching: FRAP experiments were performed by a Zeiss Zen 900 confocal microscope with x20/0.8 NA Plan- Apochromat air objective. For each of the peptides, out of a total peptide concentration of 20 mM we used 0.5% FITC-labeled peptide, in tris buffer pH 8 with 0.2 M NaCl.”**

References

1. Wang, J.; Choi, J. M.; Holehouse, A. S.; Lee, H. O.; Zhang, X. J.; Jahnel, M.; Maharana, S.; Lemaitre, R.; Pozniakovsky, A.; Drechsel, D.; Poser, I.; Pappu, R. V.; Alberti, S.; Hyman, A. A., A Molecular Grammar Governing the Driving Forces for Phase Separation of Prion-like RNA Binding Proteins. *Cell* **2018**, *174* (3), 688+.
2. Krainer, G.; Welsh, T. J.; Joseph, J. A.; Espinosa, J. R.; Wittmann, S.; de Csilléry, E.; Sridhar, A.; Toprakcioglu, Z.; Gudiškytė, G.; Czekalska, M. A., Reentrant liquid condensate phase of proteins is stabilized by hydrophobic and non-ionic interactions. *Nature communications* **2021**, *12* (1), 1-14.
3. Katzir, I.; Haimov, E.; Lampel, A., Tuning the Dynamics of Viral Factories-Inspired Compartments Formed by Peptide-RNA Liquid-Liquid Phase Separation. *Advanced Materials* **2022**, *34* (1), 2206371.
4. Sloan-Dennison, S.; Lampel, A.; Raßlenberg, E.; Ulijn, R. V.; Smith, E.; Faulds, K.; Graham, D., Elucidation of the structure of supramolecular polymorphs in peptide nanofibres using Raman spectroscopy. *Journal of Raman Spectroscopy* **2021**, *52* (6), 1108-1114.
5. Hernández, B.; Coïc, Y. M.; Pflüger, F.; Kruglik, S. G.; Ghomi, M., All characteristic Raman markers of tyrosine and tyrosinate originate from phenol ring fundamental vibrations. *Journal of Raman Spectroscopy* **2016**, *47* (2), 210-220.
6. Siamwiza, M. N.; Lord, R. C.; Chen, M. C.; Takamatsu, T.; Harada, I.; Matsuura, H.; Shimanouchi, T., Interpretation of the doublet at 850 and 830 cm⁻¹ in the Raman spectra of tyrosyl residues in proteins and certain model compounds. *Biochemistry* **1975**, *14* (22), 4870-4876.

REVIEWERS' COMMENTS

Reviewer #1 (Remarks to the Author):

While the revisions have improved the manuscript and clarified my original comments, I feel that its main message is still a bit vague and can be further improved. For example, the only sentence in the abstract (We show that the LLPS propensity...) that discusses the results is quite uninformative. I highly recommend to improve the abstracts as well as other parts in the manuscript regarding the main principles for designing and controlling LLPS of short peptides.

Most likely the ability to define molecular rules is limited by the few sequences that were studied here, yet this is the motivation of this study. Some rules are provided and are very clear. Such as the role of Trp as shown in the new sequence WGR-6/9. Additional rules or clear insights in the conclusions/abstract can improve the manuscript.

For example, a point that could deserve a better explanation is regarding the role of the ELP. It is stated that it contributes to intramolecular interactions via Arg-Tyr interactions (p 9) but in the conclusions it is stated that interactions between Tyr/Phe with Arg are crucial for droplet formation. The relationship between these intramolecular interactions and droplet formation is unclear.

Reviewer #2 (Remarks to the Author):

The authors performed many new experiments, and the manuscript has significantly improved. It will be of high interest for the field. I thank the authors for the large effort required to fully address all my comments.

Point by point responses to the reviewers' comments:

Reviewer #1

While the revisions have improved the manuscript and clarified my original comments, I feel that its main message is still a bit vague and can be further improved.

For example, the only sentence in the abstract (We show that the LLPS propensity...) that discusses the results is quite uninformative. I highly recommend to improve the abstracts as well as other parts in the manuscript regarding the main principles for designing and controlling LLPS of short peptides.

Most likely the ability to define molecular rules is limited by the few sequences that were studied here, yet this is the motivation of this study. Some rules are provided and are very clear. Such as the role of Trp as shown in the new sequence WGR-6/9. Additional rules or clear insights in the conclusions/abstract can improve the manuscript.

For example, a point that could deserve a better explanation is regarding the role of the ELP. It is stated that it contributes to intramolecular interactions via Arg-Tyr interactions (p 9) but in the conclusions it is stated that interactions between Tyr/Phe with Arg are crucial for droplet formation. The relationship between these intramolecular interactions and droplet formation is unclear.

Response: We thank the reviewer for the positive feedback. To address the reviewer's comment, we added the following text to the manuscript:

Page 15 line 6: "We show that the material properties of the droplets can be tuned by changes to the peptide sequence, where electrostatic repulsion, steric hindrance, and specific **intra-** and intermolecular interactions directly affects peptide diffusion.

Specifically, our findings suggest that intramolecular contacts between Tyr/Phe and Arg side, induced by the ELP domain, might compete with intermolecular interactions in the condensed droplet phase, resulting in accelerated diffusivity."

In addition, we revised the abstract and added the following line: "**Specifically, with the aid of Raman and NMR spectroscopy, we show that interactions between arginine and aromatic amino acids underlie droplet formation, and that both intra- and intermolecular interactions dictate droplet dynamics."**

Reviewer #2

The authors performed many new experiments, and the manuscript has significantly improved. It will be of high interest for the field. I thank the authors for the large effort required to fully address all my comments.

Response: We thank the reviewer for the positive feedback.